# Understanding Naturalistic Facial Expressions with Deep Learning and Multimodal Large Language Models

**DOI:** 10.3390/s24010126

**Published:** 2023-12-26

**Authors:** Yifan Bian, Dennis Küster, Hui Liu, Eva G. Krumhuber

**Affiliations:** 1Department of Experimental Psychology, University College London, London WC1H 0AP, UK; yifan.bian.23@ucl.ac.uk; 2Department of Mathematics and Computer Science, University of Bremen, 28359 Bremen, Germany; kuester@uni-bremen.de (D.K.); hui.liu@uni-bremen.de (H.L.)

**Keywords:** automatic facial expression recognition, naturalistic context, deep learning, multimodal large language model

## Abstract

This paper provides a comprehensive overview of affective computing systems for facial expression recognition (FER) research in naturalistic contexts. The first section presents an updated account of user-friendly FER toolboxes incorporating state-of-the-art deep learning models and elaborates on their neural architectures, datasets, and performances across domains. These sophisticated FER toolboxes can robustly address a variety of challenges encountered in the wild such as variations in illumination and head pose, which may otherwise impact recognition accuracy. The second section of this paper discusses multimodal large language models (MLLMs) and their potential applications in affective science. MLLMs exhibit human-level capabilities for FER and enable the quantification of various contextual variables to provide context-aware emotion inferences. These advancements have the potential to revolutionize current methodological approaches for studying the contextual influences on emotions, leading to the development of contextualized emotion models.

## 1. Introduction

Recent advances in computer vision have greatly facilitated the development of sophisticated affective computing systems for facial expression recognition (FER) research [1]. Researchers across domains have applied various computational techniques to analyze diverse and complex mental states, including emotions [2], pain [3], physiological correlates [4], personality traits [5], and clinical disorders [6]. Nevertheless, our understanding of facial expressions remains mostly limited to inferences drawn from laboratory studies. Facial expressions produced in controlled laboratory settings may suffer from a lack of ecological validity and fail to represent the full spectrum of facial behaviors observed in real-life scenarios [7,8]. There is a growing emphasis on investigating naturalistic facial behaviors coupled with advanced computational techniques to spark theoretical advancements in affective science [9].

Naturalistic facial expressions can be observed in connection with a wide range of psychologically significant contexts encountered in everyday situations. Naturalistic facial expressions may more accurately reflect the complex and dynamic nature of emotional experiences in the real world than expressions elicited by experimental manipulations that are often artificial or short-lived (e.g., receiving an electrical shock). One approach toward studying more ecologically valid facial expressions relies on materials sourced from third-party media such as reality shows, vlogs, movies, and documentaries. Such sources often comprise millions of facial expressions accompanied by perceptually rich contexts, which are being made available in several newly developed datasets for FER research [10]. For surveys of existing naturalistic facial expression databases, readers are referred to [11,12,13]. Examining naturalistic expressions presents vast conceptual and empirical opportunities to yield further insights into how emotions emerge and reconfigure in naturalistic situations [14], discover new emotion categories that are rarely observed in laboratory environments [15], and document a comprehensive taxonomy of facial behaviors [16].

However, there are two main challenges in the studies of naturalistic facial expressions. The first challenge relates to the difficulties in detecting and tracking facial behaviors in unconstrained environments. Naturalistic expressions are often captured in uncontrolled settings with unexpected variations in head orientation, illumination, complex background, and facial occlusions, which may result in errors in detecting facial behaviors. More robust affective computing models are required to effectively analyze facial behaviors captured in wild conditions, as traditional models often exhibit drastic drops in performance in uncontrolled environments [17]. The second challenge concerns the interpretation of naturalistic facial expressions. Unlike laboratory studies that can validate the underlying emotional experiences of expressions through self-report measurements, it is not feasible to collect retrospective data to validate naturalistic expressions sourced from the Internet. Hence, accurately inferring the emotion states of naturalistic expressions can be more challenging than working with laboratory data. Nevertheless, this issue can be addressed through multimodal annotation and a comprehensive analysis of situational contexts to specify the underlying psychological states [18].

This review is divided into two sections to discuss the applications of deep learning-based FER toolboxes and multimodal large language models (MLLMs) for tackling these challenges. The first section evaluates several newly developed and easy-to-use FER toolboxes for facial expression analysis in unconstrained environments. To support researchers in making informed decisions regarding the selection of appropriate toolboxes, we critically review the performance of five FER toolboxes, namely OpenFace 2.0 [19], Affdex 2.0 [20], Py-Feat [21], LibreFace [22], and PyAFAR [23], along with their underlying neural architectures and databases used for model training. The second section discusses the potential utilization of MLLMs for analyzing and interpreting naturalistic expressions in association with contextual cues. Naturalistic expressions can be rendered meaningful by referencing the specific contexts in which they occur and interact [18]. MLLMs such as GPT-4V [24] and LLaVA [25] exhibit promising capabilities for quantifying contextual variables, which can serve as contextualized FER models for robust, explainable emotion inferences of naturalistic expressions.

## 2. Analyzing Naturalistic Facial Expressions with Deep Learning

The general processes of facial expression analysis consist of three components: face detection, feature extraction, and the prediction of facial action units (AUs) [26] and/or emotions. In particular, feature extraction has been regarded as the most crucial component in the FER process, which can be categorized into handcrafted and learned features [15]. Handcrafted features can be extracted with methods such as histograms of oriented gradients (HOGs) and local binary patterns (LBPs) for capturing facial textures and shapes. FER models trained based on handcrafted features using shallow learning approaches (e.g., support vector machines (SVMs)) achieve good performance in facial expressions produced in laboratory settings but have recently been outperformed by models trained on learned features [27]. Learned features are directly extracted from raw data through deep neural networks (DNNs) comprising multiple layers that hierarchically learn more complex, representative spatial–temporal features than the previous layers. Accumulating research evidence suggests that DNN models consistently surpass most of the shallow learning models based on handcrafted features by a large margin [28]. Compared to shallow learning models, cross-domain experiments reveal that DNN models such as convolutional neural networks (CNNs) and vision transformers (ViTs) achieve superior generalizability and accuracy for emotion recognition [29,30,31,32] and AU detection [33,34,35] on unseen datasets with different demographics, camera views, and emotion-eliciting contexts. More importantly, deep FER models are fairly robust to variations in brightness, head poses, and occlusions [27], whereas shallow learning models trained on handcrafted features (e.g., LBPs) may be substantially impacted by variations in luminance [21] and head rotation [36].

Despite their impressive performance, accessing state-of-the-art (SOTA) deep learning models for FER research has been hindered by several obstacles. SOTA models are rarely released for public use in repositories such as GitHub or are rarely open source for end users to fine-tune the models with their own datasets for new tasks. Open source models often lack graphical user interfaces or documentation for easy implementation for users who may lack specific programming knowledge. These SOTA models might be too heavy to run for real-time analysis, restricting their potential application in practical settings. Additionally, early FER models have been primarily trained on datasets captured in controlled recording conditions with invariant illumination and fixed camera position [37]. These models may learn features that rarely align with real-world situations, which limits their effectiveness in analyzing naturalistic expressions. To bridge this gap between cutting-edge FER techniques and their implementation, we discuss a selection of the most prominent, publicly accessible, lightweight, and user-friendly toolboxes that incorporate SOTA models suitable for analyzing facial expressions in the wild.

In the remainder of this section, we will first provide an overview of two FER toolboxes that incorporate both shallow and deep learning models for facial behavior analysis, namely OpenFace 2.0 and Py-Feat. Next, we will discuss three FER toolboxes that are primarily trained on deep learning methods: Affdex 2.0, LibreFace, and PyAFAR. Table 1 summarizes the main characteristics and access information of each FER toolbox.

### 2.1. FER Toolboxes Based on Mixed Learning Models

OpenFace 2.0 [19]: OpenFace 2.0 is a representative tool of the SOTA shallow learning models based on handcrafted features (i.e., HOGs) for AU recognition. OpenFace 2.0 is capable of a variety of facial analysis tasks, including facial landmark detection, gaze and pose estimation, and AU detection. OpenFace 2.0 uses a deep convolutional expert-constrained local model (CE-CLM) for facial landmark detection and tracking, which is trained on wild datasets with nonfrontal faces and varying illumination. It can better detect profiles or severely occluded faces than its previous version [37]. For AU detection, OpenFace 2.0 relies on shallow learning algorithms, including linear SVM for binary detection (presence or absence) and support vector regression (SVR) for intensity probability estimation. The AU models are trained in seven laboratory databases with AU annotations, which contain posed (Bosphorus [38]; FERA 2011 [39]), spontaneous (CK+, also containing posed expressions [40]; UNBC-McMaster [3]; DISFA [41]; and BP4D/FERA 2015 [42]), and conversational (SEMAINE [43]) expressions. Research by [36] demonstrated that OpenFace 2.0 outperformed several commercial FER toolboxes such as FaceReader 7.0 [44] for AU detection on datasets that contain posed (DISFA+ [45]), conversational (GFT [46]), and wild (Aff-wild2 [47]) expressions. However, OpenFace 2.0 was found to have inferior generalizability ability in AU detection on unseen datasets compared to deep FER models [48]. This is likely due to the limited discriminative values of handcrafted features or the inability of shallow learning algorithms to capture intricate, nonlinear patterns of facial behaviors.

Py-Feat [21]: Py-Feat provides various pretrained models based on both handcrafted and learned features, allowing users to flexibly decide which combinations of models to use according to specific task requirements. Py-Feat includes several face detection models such as multitask convolutional neural networks (MTCNNs [49]) and RetinaFace [50], which demonstrate robustness on partially obscured or nonfrontal faces. For AU detection, Py-Feat uses popular shallow learning methods for computing binary results with SVM and continuous results with optimized gradient boosting (XGB [51]), which are pretrained on both laboratory (BP4D, DISFA, CK+, and UNBC-McMaster) and wild (Aff-wild2 [47]) datasets. Its AU models have been shown to be slightly less accurate than the reported results of OpenFace 2.0 benchmarking on the DISFA+ dataset. However, Py-Feat yielded more consistent and reliable AU estimation for faces with varying head orientations, whereas the performance of OpenFace 2.0 drops dramatically when head angles are larger than 45 degrees [36]. The superior performance of Py-Feat on AU detection for occluded faces might be attributed to the inclusion of the wild dataset [47], which contains nonstandardized facial images and videos for pretraining models. Py-Feat provides direct emotion inferences for six basic emotions with emotion models trained on datasets with spontaneous (CK+ [40]), posed (JAFFE [52]), and wild (ExpW [53]) expressions. Since not all facial regions contribute equally to emotion perception, and some facial regions may be occluded in unconstrained conditions, Py-Feat exploits the residual masking network (ResMasNet [54]), a deep learning method that utilizes attention mechanisms to adaptively weight and select the most emotionally salient regions of the face while ignoring irrelevant features (e.g., facial occlusion). The ResMasNet outperforms another shallow learning model (i.e., SVM) adopted in Py-Feat and a commercial FER toolbox (FACET [55]) on a wild dataset (AffectNet [56]). Py-Feat provides numerous functional packages for data preprocessing, statistical analyses (e.g., time-series correlation and regressions), and visualization, which facilitate data exploration. Py-Feat is written in the Python programming language. Although a graphical user interface is currently not available in Py-Feat, it is relatively easy to use following step-by-step tutorials.

### 2.2. Deep FER Toolboxes

Affdex 2.0 [20]: Affdex 2.0 is a commercial software program designed to analyze facial behaviors in the wild. For face detection, it exploits region-based convolutional neural networks (R-CNNs [57]), which perform better on challenging conditions (e.g., variations in illumination, hand occlusions, etc.) compared to Affdex 1.0 [58]. For AU detection, Affdex 2.0 utilizes large samples of a private dataset collected using a web-based approach [59] to train CNN models, which showed enhanced accuracy and less biased results on ethnic minorities (e.g., African and South Asian) than its previous version. On benchmark evaluation for AU detection, Affdex 2.0 outperforms numerous SOTA methods on the DISFA dataset, which contains spontaneous expressions induced by emotion-evoking videos. Although Affdex 2.0 performs slightly worse than OpenFace 2.0, it is important to note that the AU models of Affdex 2.0 have not been trained on the DISFA, which may bias the result. Unlike other FER toolboxes that directly predict emotions with separate models specifically trained on emotional expression datasets, Affdex 2.0 estimates seven basic emotions [60] based on the reverse inference of the activation of AUs with assigned weights postulated in EMFACS [61]. Negative weights are assigned for emotion prediction when opposite AUs occur to reduce a false-positive rate. For instance, the presence of a “lip corner lowerer” (AU15) may decrease the likelihood of predicting “happiness” based on the activation of a “lip corner raiser” (AU12). Affdex 2.0 outperforms several SOTA emotion models on the Aff-wild2 dataset [47]. In addition to basic emotions, Affdex 2.0 can predict other affective states such as confusion, sentiment, and engagement based on predefined rules for AU activation. Although it is a commercial software program, Affdex 2.0 has been trained on a large sample of wild dataset with 11 million annotated images of people of all ages, genders, and ethnicities, which might be otherwise difficult to collect and annotate without financial support. By contrast, many open source systems are trained on a comparatively limited number of publicly accessible laboratory datasets with a small number of participants and a narrow range of demographic diversity. They are thus at a disadvantage with respect to the development of robust, unbiased models. However, since Affdex 2.0 is not open source, it does not allow users to further fine-tune the models for downstream tasks.

LibreFace [22]: LibreFace is a newly developed toolkit that incorporates several SOTA deep networks for facial AU and emotion expression analysis. LibreFace utilizes MediaPipe [62] for precise face detection with 468-point 3D landmark registration and normalization with geometric transformation. Model performances including feature extraction, robustness, and generalizability are enhanced by several pretraining processes. Specifically, a ViT-base model and a ResNet-18 model [63] were pretrained on several large wild datasets sourced from the Internet, including the training set of AffectNet [56], EmotioNet [64], and FFHQ [65], which consist of millions of facial images with a wide range of variations in demographic features, illumination, and head orientation. After the pretraining phase, the models were fine-tuned on the DISFA dataset [41] for AU intensity estimation and the BP4D-Spontaneous datasets [66] for AU detection. LibreFace leverages the Swin Transformer model [67] to capture the spatial correlations and interactions between different facial features from a global perspective, which could achieve better performance than traditional CNNs that focus on local patterns and features within facial subregions [68,69]. Moreover, LibreFace bypasses the need for large, labeled datasets by exploiting the masked autoencoder (MAE) method that allows for learning representative facial features through the process of image and video reconstruction [70]. LibreFace utilizes feature-wise knowledge distillation to reduce computational costs, thereby boosting the inference efficiency for real-time facial expression analysis. LibreFace outperforms OpenFace 2.0 [19] and other SOTA deep learning models [35,71] on the DISFA and BP4D datasets for predicting the activation of AU. For emotion prediction, LibreFace achieves competitive results comparable or superior to more complicated and heavier SOTA models [72] on two wild datasets, AffectNet and RaF-DB [73]. LibreFace is currently open source in Python, and for Windows users, an easy-to-use graphical user interface is available using the OpenSense platform [74].

PyAFAR [23]: PyAFAR is developed for facial AU detection and intensity estimation in addition to head orientation and facial landmark detection. For face detection, it uses MediaPipe and Facenet [75], which can identify and track individuals even if they exit and re-enter the field. PyAFAR adopts two separate DNN models based on ResNet50 [63], with increased depth of neural architecture to perform more effective convolution operations for complex facial feature representation. The models have been pretrained on the ImageNet [76] dataset for detecting 12 Aus in adults and 9 Aus in infants. The adult model is fine-tuned on BP4D+ [77], an expansion of the BP4D-Spontaneous dataset containing spontaneous facial expressions induced by both active (e.g., singing) and passive (e.g., watching emotionally loaded videos) tasks. The infant model is trained on the MIAMI [78] and CLOCK [79] databases, which capture infants’ responses induced by experimental procedures, such as the removal of attractive toys and still face paradigms [80]. Both adult and infant models achieve accurate results on within-domain validation [23]. PyAFAR shows superior cross-database performance on the GFT dataset [46] compared to OpenFace 2.0 and the previous version of AFAR [81]. An executable interface and a step-by-step visual instruction guide are available for the easy implementation of the toolbox.

These FER toolboxes empower researchers to effectively address the challenge of unexpected variation in naturalistic behaviors acquired from unconstrained environments. However, it is important to note that some FER toolboxes (e.g., Affdex 2.0, LibreFace, and Py-AFAR) have only been recently created and validated by their developers. Further empirical research conducted by independent researchers is required to compare and validate the cross-domain performance of these toolboxes [1].

What is still missing in the studies of naturalistic facial expressions is the lack of a comprehensive analysis of contextual information critical for a naturalistic understanding of emotions [82]. Prior works have focused extensively on the analysis of facial features using FER toolboxes for emotion inferences, while contextual variables have been largely ignored [83]. Emotion inferences made solely based on decontextualized faces are ecologically invalid and meaningless. For instance, a smile can be reliably recognized as expressing “happiness” by FER toolboxes, but it is difficult to evaluate the meaning of the emotion without referencing the emotional stimuli or surrounding contexts (e.g., smile as reflecting the anticipation of a music festival [84,85]). Elucidating the interaction between facial behaviors and concurrent contexts is an important research question for affective science.

A comprehensive analysis of contextual elements can provide important cues for an accurate assessment of the underlying emotional experience associated with naturalistic facial expressions [18]. For example, naturalistic facial behaviors (e.g., a smile) are often accompanied by contextual cues presented in various forms such as clothing (e.g., a gown), scenery (e.g., wedding venues), activities (e.g., marriage proposal), voices (e.g., “I love you”), body postures (e.g., holding hands), other faces, and so forth, which shape how faces are perceived (e.g., the enjoyment of interpersonal connection). Contextual variables can be measured and quantified by human annotators [86]. When facial expressions are presented with perceptually rich contextual information, human annotators show substantially greater agreement for labeling facial expressions than decontextualized faces [87]. This indicates that the current limitations of evaluating facial expressions with FER systems could be addressed by including contextual cues, as human perceivers can make more robust, reliable emotion inferences. However, annotating naturalistic expressions and their contexts can be more complicated and labor-intensive than laboratory datasets. Advanced multimodal annotation tools [88] may help provide multimodal annotation to evaluate facial expressions together with other nonverbal modalities and rich contextual information to provide accurate portrayals of the interaction between facial expressions and contexts. In the following section, we discuss the novel applications of MLLMs that could circumvent the need for extensively annotated datasets, fostering further advancement in naturalistic affective research.

## 3. Advancing Naturalistic Affective Science with Multimodal Large Language Models

Recent advancements in MLLMs have demonstrated remarkable versatility and capability in various domains and tasks. Although MLLMs are not specifically programmed for emotion recognition tasks, such capabilities emerge as the result of data scaling [89]. The main idea of using MLLMs for emotion recognition is to use powerful large language models as an intelligent brain to process and align textual, visual [25,90], and/or auditory [91,92] information to perform emotion inferences [93]. There is an increasing number of open source MLLMs, including LLaVA [25] and MiniGPT-4 [90], available on platforms such as HuggingFace. Many MLLMs also provide a user-friendly web interface that enables more flexible interactions with the user. This section provides an in-depth discussion on the emergent novel applications of MLLMs for context-aware emotion recognition, generalizable facial expression analysis, and adaptability to other related tasks such as the classification of nuanced emotion categories.

### 3.1. MLLMs as a Contextualized Emotion Model

MLLMs can serve as a contextualized model quantifying contextual variables to provide robust emotional reasoning for naturalistic expressions (see Figure 1 for example). Several studies have demonstrated the exceptional capabilities of MLLMs in identifying emotionally evoking context and comprehending how these contextual cues may influence the emotional state of a person [94]. To quantify contextual variability, MLLMs can be used to perform a wide range of visual reasoning tasks such as spatial relationship analysis and object recognition to understand the visual world with a simple prompt, i.e., “describe the image”. This can provide a detailed description of the situational contexts, which can then be used to infer relevant emotional states and related antecedents or consequences. Such contextualized inferences are more in line with how humans naturally perceive emotions in real-life situations by synthesizing concurrent multisensory information from the face and contextual cues [95]. This potential application is exemplified in a recent technical report of GPT-4V in the sense that it can make accurate emotion inferences based on integrated contextual cues such as “protest crowd” and “presence of policies” for inferring “injustice” and “anger” [24]. Moreover, research by Etesam et al. [94] has revealed that MLLMs outperformed vision language models in context-aware emotion recognition tasks using the EMOTIC dataset [96], which contains contextually rich images annotated with 26 emotion categories. While vision language models like CLIP [97] are effective at detecting immediately visible characteristics such as facial expressions, body postures, and activities, they fail to reason about the underlying causal relationships of these contextual data for emotion inferences. In contrast, MLLMs like LLaVA [25] not only identify these visible characteristics but also integrate and capture the complex relationships among these contextual cues for emotion inferences. For instance, while CLIP may perceive “raising arms” as signifying “surprise” and “fear”, MLLMs may reason that this body posture actually reflects “happiness” and “excitement” given the context of skiing. This empirical evidence supports the practical utility of MLLMs for affective research, which can further improve our understanding of how naturalistic emotions manifest in real-life scenarios by considering contextual variability.

Furthermore, the contextual reasoning generated from MLLMs can complement the results obtained from FER toolboxes to produce context-aware emotion inferences, which are more robust and insightful than simply analyzing the face alone. This is particularly important for accurately identifying and interpreting complex or vague expressions that convey mixed emotional signals such as sarcasm and Schadenfreude indicated by the incongruence between facial behaviors and contexts (e.g., a polite smile accompanied by sarcastic statements [98]). More importantly, future studies can utilize MLLMs and FER toolboxes for examining the relationship between different types of contexts and facial expressions. Past research by Cowen et al. [14] has utilized sophisticated DNNs to automatically classify facial expressions and contexts from over millions of videos sourced from the Internet. They found that sixteen types of facial expressions consistently occur in specific contexts, suggesting substantial contextual dependence on facial expressions. Nevertheless, in this study, the DNN models for context classification could only predict the topics of videos. For instance, a context like “the bride dressed in a gown dancing with the groom dressed in a suit in a banquet hall filled with tables and wine glasses” may be simplified into a label like “wedding”. This may lead to the oversimplification of contextual variability that could significantly alter emotion perception. A small object shown in the wedding context, such as a gun, may completely shift the emotion perception from “joy” to “fear”. Therefore, future studies are encouraged to utilize MLLMs to generate detailed context descriptions, as illustrated in Figure 1, to fully examine the complex relationships between facial expressions and contextual variability. In this way, researchers can not only leverage contextual information for inferring the emotional states of naturalistic expressions but also utilize facial information to guide the prediction of situational contexts (e.g., appeasement smile indicating submissive contexts [99]).

**Figure 1 sensors-24-00126-f001:**
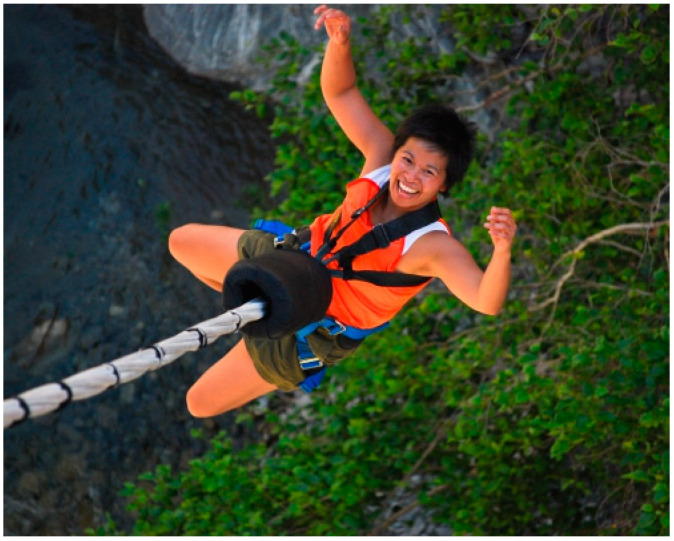
Contextualized emotion inferences generated from LLaVA-7B [25] (image from OASIS [100]). Temperature is set to zero for reproducible response; note that the model can identify the association between the emotion of “excitement” and the activity of “swing through the air”, but it mistakenly describes the person as wearing a black helmet. Prompt: “Describe the image? What emotions can be inferred based on the context?”. LLaVA-7B: “The image features a woman wearing an orange vest and a black helmet, hanging from a rope or wire. She appears to be enjoying herself as she swings through the air, possibly on a rope swing or a zip line. The woman’s smile and the fact that she is wearing a helmet suggest that she is having a fun and adventurous time. The scene conveys a sense of excitement and joy, as the woman embraces the thrill of the activity.”

### 3.2. MLLMs as Generalizable FER Toolboxes

Several technical reports have documented the impressive human-level capabilities of MLLMs for analyzing facial expressions [24]. Researchers can conduct FER tasks using MLLMs with a simple query “What is the facial expression?” MLLMs can detect emotions from facial expressions and can generalize well to unseen data because MLLMs are developed to address a diverse range of tasks. MLLMs can also analyze a wide spectrum of emotion categories based on natural language processing, which allows researchers to capture more nuanced emotions (e.g., awe, shame, emotional suppression, etc. [24]). By contrast, prior emotion models have typically been programmed to predict a limited set of basic emotions, which restrict their potential applications in more generalized domains (e.g., monitoring student engagement in the classroom or pain in the ICU). MLLMs can also provide multiple emotion labels for comprehensive descriptions of complex emotional states. Such responses are more aligned with the human perception of facial expressions that have been conceptualized as a manifold representation of varying degrees of emotions (e.g., a face may be perceived as 51% anger and 49% disgust [101]). In comparison, most existing emotion models rely on majority voting to determine the single most representative label [102]. Although such approaches may enhance the overall reliability of emotion prediction, they may exclude other relevant but less dominant emotions and fail to detect compound emotions [103]. Moreover, unlike prior works that mainly focus on emotion prediction, MLLMs can take a further step to provide reasonable explanations of the facial features being used for emotion inferences (see Figure 2, for example). For instance, after inferring a facial image as showing “fear” or “anxiety”, GPT-4V can offer detailed descriptions of the visual cues to interpret these emotional states (e.g., “wide-open eyes” revealing “a sense of shock or surprise” and “hands covering the mouth” indicating “suppressing scream” [24]). It can also modify its emotional inferences based on hand gestures, which are often ignored or treated as facial occlusion by existing FER toolboxes. Furthermore, Lian et al. [93] examined the performance of several MLLMs, including Video-LLaMA [91], PandaGPT [92], and Valley [104], for generating explainable emotional inferences from a subset of the MER2023 dataset [105] comprising dynamic facial expressions sourced from the Internet. Specifically, they evaluated the abilities of MLLMs to identify emotion-related cues (e.g., “lifted eyebrows”, “smiling face”, etc.) and predict emotional states (e.g., “happiness”) based on the identified cues. In addition, the plausibility of the emotion reasoning process was validated by human observers. Valley consistently achieved the best performance, with 72.1% accuracy in identifying emotion-related cues and 57.8% accuracy in predicting emotional states. It should be noted that the paper did not report the base rates for emotion prediction, and the authors used ChatGPT to relabel more subtle emotions, going beyond the original labels introduced in MER2023 as a baseline. The emotion reasoning process demonstrated by Valley also aligned most closely with human perception, receiving a plausibility score of 65.0%. This study provides empirical evidence demonstrating the capabilities of MLLMs in identifying explainable cues for emotion inferences, which supports their potential utility in FER research. However, further efforts are required to improve the model’s performance. With the enhanced interpretability of emotion recognition processes, FER researchers can be more confident in asserting the psychological states of naturalistic expressions, thereby achieving greater reliability and specificity.

### 3.3. Adaptability of MLLMs for Different Emotion Recognition Tasks

MLLMs have strong adaptation capabilities for more challenging emotion recognition tasks through few-shot in-context learning (ICL). Few-shot ICL refers to the ability to quickly adapt to novel tasks given a short instruction and few examples without fine-tuning the models and abundant labeled data. After few-shot ICL, MLLMs such as Flamingo [106] can outperform some of the SOTA contrastive models [97] despite using only around 30 task-specific examples, which are about 1000 times fewer data inputs required for fine-tuning the models. There are numerous potential applications of few-shot ICL for naturalistic affective research such as the classification of nuanced emotion categories. For instance, several large-scale datasets contain a broad spectrum of facial behaviors captured in the wild, which present new opportunities for investigating the complexity and variability of emotional experiences and their underlying psychosocial processes in real-life scenarios. However, many of these datasets are only annotated with a few emotion classes, which limits their potential utilization in affective research. With the advanced ICL functionality, researchers can further exploit these datasets by applying MLLMs to identify new emotion categories for a more fine-grained analysis of human emotions [93]. Specifically, FER researchers can provide a few demonstration examples of facial expressions categorized by specific emotions in the format of image–test or video–text pairs to extrapolate to new emotion recognition tasks with a visual query such as “identify the images that display the same facial expression illustrated in the above examples”. This approach may enable FER researchers to evaluate more specific and contextualized expressions in the wild that may not be accounted for by the limited set of expressions detected by existing FER toolboxes. This could also pave the way for examining the assumptions of various emotion theories, e.g., appraisal theories [107], the theory of constructed emotions [82], and the behavioral ecology view [108]. Coupled with ICL techniques, researchers can also examine if facial expressions produced in laboratory settings generalize to naturalistic contexts, thereby gaining deeper insights into the ecological validity of these facial displays and their potential implications in real-world situations. Unfortunately, to the best of our knowledge, in-context learning tailored specifically for FER tasks has not received much empirical attention. Future studies should leverage the intriguing emergent ability to further advance the field of naturalistic affective science.

### 3.4. Limitations of MLLMs

While MLLMs have the potential to revolutionize the domain of FER research, it is essential to acknowledge their limitations and work toward addressing them. For facial behavior analysis, it is unclear whether MLLMs can provide FACS-like inferences [26] that are precise enough for accurate facial behavior analysis. It is important to test if MLLMs can capture the variations in facial parameters [109] and distinguish the subtle differences between various types of facial behaviors [110]. For instance, smiles can be characterized by different facial configurations such as Duchenne smiles (e.g., AU6 + 12), “selfie smiles” (e.g., AU13), or “miserable” smiles (e.g., AU12 + 14 or 12 + 15), which are associated with distinct psychological states [111]. Further empirical investigation is required to systematically examine the similarity between the facial behaviors described by MLLMs and the facial AUs detected by FER toolboxes, as well as explore methods to fine-tune the models to achieve comparable or superior results. Before such empirical testing is carried out, it is recommended to incorporate both MLLMs and FER toolboxes for fine-grained facial behavior analyses. In addition, it remains unclear to what extent the contextual perception of MLLMs aligns with human perception in terms of emotion inferences [93,112]. It is possible that contextualized emotion inferences may be biased by random noise in the context. Variability also exists in the susceptibility to contextual influences among individuals [113], as evidenced in the case of depressed individuals who often exhibit facial responses insensitive to contextual cues [114]. Therefore, it is critical to formulate theories and models to elucidate the mechanisms underlying the integration of contextual information during the process of facial expression recognition [115]. Finally, it is crucial to acknowledge that despite an extensive search for empirical evidence across various domains, some arguments concerning MLLMs remain hypothetical, particularly regarding their adaptability, as discussed in Section 3.3. Although MLLMs have demonstrated promising capabilities in addressing numerous important research questions in affective science, as illustrated in Section 3.1 and Section 3.2, they are still in the early phases of development, awaiting further improvement. Further empirical research is required to explore the versatility of MLLMs for emotion recognition tasks, thereby bridging the gap between their potential applications and practical implementation.

## 4. Conclusions

In this review, we have provided an overview of publicly accessible and user-friendly FER toolboxes for robust facial analyses under unconstrained conditions and an introduction to the potential applications of MLLMs to further advance the field of affective science. However, it is important to emphasize that most of these techniques have only been recently developed. Additional empirical investigations are needed to validate their practical utility across various domains and further improve their performance. Future studies are encouraged to utilize these sophisticated techniques to expand our knowledge of naturalistic facial expressions and develop contextualized emotion models to achieve a comprehensive understanding of emotional experiences in the real world [116].

## Figures and Tables

**Figure 2 sensors-24-00126-f002:**
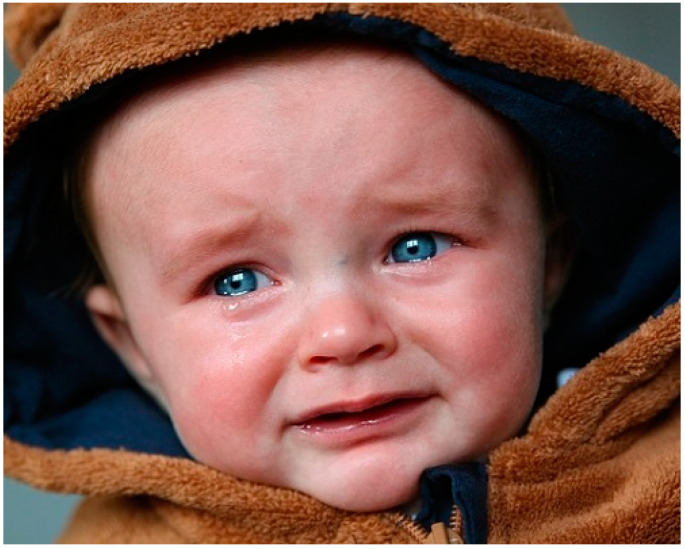
Explainable facial expression recognition generated from LLaVA-7B [25] (image from OASIS [100]). Temperature is set to zero for reproducible response. Prompt: “What emotions are shown on the baby’s face? What specific facial cues are used to infer the emotions?” LLaVA-7B: “The baby’s face shows a mix of emotions, including sadness, fear, and possibly discomfort. The baby’s facial cues include a frowning expression, tears in the eyes, and a slightly puckered mouth. These cues suggest that the baby is experiencing negative emotions or discomfort, possibly due to the situation or the environment around them”.

**Table 1 sensors-24-00126-t001:** FER toolbox comparison on functionality, neural architecture, and type of dataset used for training emotion or AU models. The deep learning models are shown in bold. The datasets are categorized as W (wild), S (spontaneous), or P (posed), representing facial expressions sourced from the Internet or nonlaboratory environments, induced by experimental procedures, or deliberately mimicked by actors in laboratory settings, respectively. * Please consult the website for complete documentation of all face detection models incorporated in Py-Feat. ** The emotion model of Affdex 2.0 is not specified as it is based on the activation of AUs. *** The dataset used to train AU models in Affdex 2.0 is considered spontaneous despite being captured in nonlaboratory settings.

	OpenFace 2.0	Py-Feat	Affdex 2.0	LibreFace	PyAFAR
Face Detection	**CE-CLM**	**MTCNN**, **RetinaFace** *	**R-CNNs**	**MediaPipe**	**MediaPipe**, **Facenet**
Emotion Recognition		**ResMasNet**, SVM	**	**ViT, ResNet-18**	
Action Unit	SVM, SVR	SVM, XGB	**CNNs**	**ViT, ResNet-18**	**ResNet-50**
Datasets	P, S	P, S, W	S ***	S, W	S
Open Source/Free	Yes	Yes		Yes	Yes
Graphical User Interface	Yes		Yes	Yes	Yes
Website	github.com/TadasBaltrusaitis/OpenFace (accessed on 29 November 2023)	py-feat.org (accessed on 29 November 2023)	www.affectiva.com (accessed on 29 November 2023)	github.com/ihp-lab/LibreFace (accessed on 29 November 2023)	affectanalysisgroup.github.io/PyAFAR2023 (accessed on 29 November 2023)

## Data Availability

No new data were created or analyzed in this study. Data sharing is not applicable to this article.

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
