# Peer review of "Understanding Naturalistic Facial Expressions with Deep Learning and Multimodal Large Language Models"

_sensors, 2023, doi:10.3390/s24010126_

Round 1
Reviewer 1 Report
Comments and Suggestions for Authors
The article titled "Understanding Naturalistic Facial Expressions with Deep Learning and Multimodal Large Language Models" analyzes the affective computing systems for facial expression recognition (FER) in natural contexts. It reviews user-friendly FER toolboxes incorporating advanced deep learning models and discusses their architecture, datasets, and performance. The paper also covers multimodal large language models (MLLMs) and their applications in affective science, highlighting their role in context-aware emotion inference. I would appreciate further clarifications and enhancements on the following aspects:
1. Introduction:
It would be beneficial if the authors could consider incorporating an overview of publicly available Naturalistic Facial Expressions datasets and their classification categories to provide readers with a comprehensive understanding of the landscape.
2. Analyzing Naturalistic Facial Expressions with Deep Learning:
l I suggest reconsidering the placement of the introduction of OpenFace 2.0, which is based on shallow learning models, in a section titled "Analyzing Naturalistic Facial Expressions with Deep Learning." This consideration aims to prevent potential confusion.
l In section 2.1, it might be more engaging for readers if, instead of solely describing various toolboxes, the authors could perform a comparative analysis of recognition results using various toolboxes on publicly available Naturalistic Facial Expressions datasets or private datasets. This approach would offer readers a more practical and interesting perspective.
l Consider citing and summarizing existing literature on the recognition rates achieved by the mentioned toolboxes to strengthen the textual descriptions.
3. Advancing Naturalistic Affective Science with Multimodal Large Language Models:
To offer a more comprehensive understanding, I suggest encouraging the authors to discuss the outcomes of experiments conducted across a larger dataset or various scenarios. Including a more in-depth analysis derived from bulk data usage would enhance the comprehensiveness of the review.
4. Limitations of MLLMs:
It would be beneficial for the authors to consider including practical experiments and real-world findings in the analysis of MLLMs. For instance, in lines 386-388, I suggest referencing existing studies or experimental results to support the conclusion rather than using ambiguous language.
5. Conclusion:
In the conclusion, it would be valuable to emphasize the need for further empirical research and practical experiments to validate the findings and overcome the limitations identified.
6. References:
Please verify the correctness of the references, particularly reference 4 on line 28.
7. Formatting:
It would be advisable to ensure consistency in the use of quotation marks throughout the manuscript (e.g., line 249 - double quotes; line 299 - single quotes).
8. Section Numbering:
Rectify the duplication of section numbering (Section 3.2 appears twice).
Author Response
- “It would be beneficial if the authors could consider incorporating an overview of publicly available Naturalistic Facial Expressions datasets and their classification categories to provide readers with a comprehensive understanding of the landscape.”
Author Response:
While we appreciate this suggestion, we are concerned that an overview of naturalistic FE datasets goes beyond the scope of this paper. For surveys of existing naturalistic databases (although not exhaustive), we now refer the readers to works done by Siddiqui et al. (2022) and other researchers in the manuscript (p.2, lines 44-45). Due to limitations in space we decided to focus on conceptual aspects of naturalistic facial expressions that are crucial for understanding the topic. We explain the complexity and ecological validity of naturalistic facial expressions compared to those produced in laboratory settings, as well as the theoretical advances that can be achieved by investigating naturalistic facial expressions. It is worth noting that naturalistic facial expression databases sourced from the Internet can exhibit significant variations. A separate review on this topic is necessary to thoroughly document characteristics of these databases. Moreover, we believe it is important to prioritize discussion on the available toolboxes rather than the datasets. Only with the right tools can we maximize the value of these datasets, unlocking their full potential for valuable insights and discoveries.
- “I suggest reconsidering the placement of the introduction of OpenFace 2.0, which is based on shallow learning models, in a section titled "Analyzing Naturalistic Facial Expressions with Deep Learning." This consideration aims to prevent potential confusion.”
Author Response:
We thank the reviewer for sharing this concern. To avoid potential confusion, we have now created a separate subsection titled ‘2.2 FER Toolboxes based on Mixed Learning Models’ (p.3) that place OpenFace 2.0 and Py-Feat together, which incorporate both shallow and deep learning models. Because the discussions of FER toolboxes are re-ordered, we have slightly modified the content about the comparison between Affdex 2.0 and other toolboxes (p4, lines 180-182 & 189-190)
- “In section 2.1, it might be more engaging for readers if, instead of solely describing various toolboxes, the authors could perform a comparative analysis of recognition results using various toolboxes on publicly available Naturalistic Facial Expressions datasets or private datasets. This approach would offer readers a more practical and interesting perspective..”
Author Response:
While we greatly appreciate this advice, conducting empirical testing to further validate each FER toolbox is beyond the scope of this review. However, we intend to conduct comprehensive empirical testing on these toolboxes as part of our future work.
- “Consider citing and summarizing existing literature on the recognition rates achieved by the mentioned toolboxes to strengthen the textual descriptions”
Author Response:
We noticed that different studies have adopted various metrics to evaluate the performance of the toolboxes, making it difficult to directly compare the performance of these toolboxes. We have also checked all the works that cited these FER toolboxes (except for OpenFace 2.0, which has over a thousand citations), but we did not find additional research works that perform comparative analysis of these toolboxes. Because Affdex 2.0, LibreFace, and Py-AFAR have only been recently introduced, there is also relatively limited empirical testing on these toolboxes. So far, they have only been validated by their developers or compared with the previous version. Hence, it might not be feasible to report their recognition rates. We have now included additional text (p. 6, lines 249-252) for cautioning readers about this issue and encouraging further empirical validation efforts on the cross-domain performance of these toolboxes by independent researchers.
- “Advancing Naturalistic Affective Science with Multimodal Large Language Models: To offer a more comprehensive understanding, I suggest encouraging the authors to discuss the outcomes of experiments conducted across a larger dataset or various scenarios. Including a more in-depth analysis derived from bulk data usage would enhance the comprehensiveness of the review.”
Author Response:
Empirical studies that test the performance of MLLMs on large datasets are scarce, if they even exist. When re-analysing the literature, we were fortunate to identify new empirical work by Etesam et al (2023; published on Oct 30th). We have now incorporated the findings from this study into our discussion in subsection 3.1 (p.7, lines 311-324) to support our claims. However, it is worth noting that the majority of existing works still rely on deep learning techniques to investigate naturalistic expressions. Considering this, we now discuss in greater detail (p.8, lines 333-345) the limitations of past research that attempted to use DNNs to examine naturalistic expressions from a large dataset and how that limitations can be potentially addressed by MLLMs. By presenting these drawbacks and highlighting how MLLMs can address them, we hope to encourage researchers to adopt MLLMs to advance the field. Additionally, we have included a discussion on an experiment conducted by Lian et al. (2023) to support our claim in subsection 3.2 (p.9, lines 373-387). We believe this review paper is the first to highlight the need for empirical testing in this area. Having said that, this endeavor would require significant computational resources which many research labs (including our own) may not possess.
- “Limitations of MLLMs: It would be beneficial for the authors to consider including practical experiments and real-world findings in the analysis of MLLMs. For instance, in lines 386-388, I suggest referencing existing studies or experimental results to support the conclusion rather than using ambiguous language.”
Author Response:
Once again, due to the scarcity of empirical testing on MLLMs, we were unable to identify additional evidence to further our claims. We also could not conduct experiments due to limited computational resources in our laboratory to support the claim about whether MLLMs can make FACS-like inferences (p.11). Nevertheless, we believe it is an important research question worthy of empirical investigation. Therefore, we have modified the content such that limitations are now described as an open issue for future investigation (p.11, lines 438-448).
- “In the conclusion, it would be valuable to emphasize the need for further empirical research and practical experiments to validate the findings and overcome the limitations identified.”
Author Response
We appreciate the reviewer for offering this advice. We have now included additional text to acknowledge the limitation in subsection 3.4 (p.11, lines 456 to 464) and in the conclusion (p.11, lines 469-471) to highlight the necessity for additional empirical research to support the practical utility of FER toolboxes and MLLMs in affective science.
- “References: Please verify the correctness of the references, particularly reference 4 on line 28.”
Author Response
We have re-checked the references to ensure all cited references are indeed appropriate.
- “Formatting: It would be advisable to ensure consistency in the use of quotation marks throughout the manuscript (e.g., line 249 - double quotes; line 299 - single quotes).”
Author Response
Thanks for this attentive advice. We have now used double quotes consistently throughout the paper.
- “Section Numbering: Rectify the duplication of section numbering (Section 3.2 appears twice)”
Thanks again for pointing out the formatting issue. We have corrected the section numbering issue.
Reviewer 2 Report
Comments and Suggestions for Authors
The submitted review manuscript proposes a comprehensive overview of systems for studying Facial Expression Recognition (FER) in naturalistic contexts. This study introduces information on FER toolkits and discusses the potential application of multimodal large language models (MLLMs) in emotion detection. The integration of this information has the potential to create new methodological approaches for studying the contextual influence on emotions, enabling the quantification of various contextual variables to provide context-aware emotion inferences. Such an approach could lead to the development of contextualized emotion models. The work is likely to be of interest to researchers in these subject domains.
1. The structure of the review is clear and appropriate for articles of this type.
2. The list of references is exhaustive and up to date.
3. The work provides an overview of publicly available and user-friendly FER toolkits for reliable facial analysis in unrestricted conditions.
4. The potential application of MLLMs for further development in the domain of affective science is also presented.
It is worth noting that combining these two approaches (# 3 and 4) could expand the understanding of naturalistic emotions, considering contextual variables in facial expression studies.
As a goal of the study, the manuscript recommends using the discussed methods to develop contextualized emotion models to achieve a comprehensive understanding of emotional experiences in the real world.
In the expert's opinion, the manuscript can be accepted for publication as it is.
Author Response
We are thankful for this supportive comment. No modification is made based on this comment.
Reviewer 3 Report
Comments and Suggestions for Authors
1. What sensors do you use to recognize facial emotions?
2. What is new in your article? What method did you develop?
3. If you are afraid of your experiments, bring them so that you can be sure of the validity of the judgments of your method
4. How to make sure that the emotions you recognized are true? Bring the evidence.
5. Why did you give an article in Sensors magazine?
6 Conclusions are very short
Author Response
- What sensors do you use to recognize facial emotions? 5. Why did you give an article in Sensors magazine?
Facial expressions can be detected by computer vision techniques, which rely on the utilization of biooptical sensors. Research on computer vision-based facial expression has gained significant popularity, evident from its frequent coverage in the journal Sensors. Hence, we believe it is suitable to submit our paper to this journal. Below we list some of the relevant articles that have previously been published in the same journal:
- Othman, E., Werner, P., Saxen, F., Al-Hamadi, A., Gruss, S., & Walter, S. (2021). Automatic vs. Human recognition of pain intensity from facial expression on the x-ite pain database. Sensors, 21(9), 1–19. https://doi.org/10.3390/s21093273
- Guerdelli, H., Ferrari, C., Barhoumi, W., Ghazouani, H., & Berretti, S. (2022). Macro-and Micro-Expressions Facial Datasets: A Survey. Sensors, 22(4), 1–34. https://doi.org/10.3390/s22041524
- Gavrilescu, M., & Vizireanu, N. (2019). Predicting depression, anxiety, and stress levels from videos using the facial action coding system. In Sensors (Switzerland) (Vol. 19, Issue 17). https://doi.org/10.3390/s19173693
- Namba, S., Sato, W., Osumi, M., & Shimokawa, K. (2021). Assessing automated facial action unit detection systems for analyzing cross-domain facial expression databases. Sensors, 21(12). https://doi.org/10.3390/s21124222
- Rathod, M., Dalvi, C., Kaur, K., Patil, S., Gite, S., Kamat, P., Kotecha, K., Abraham, A., & Gabralla, L. A. (2022). Kids’ Emotion Recognition Using Various Deep-Learning Models with Explainable AI. Sensors, 22(20). https://doi.org/10.3390/s22208066
- What is new in your article? What method did you develop?
Please note that our paper is a review article, not an empirical study. Rather than conducting experiments to develop and test new methods, our aim is to contribute to the field of FER research by synthesizing existing knowledge and offering new perspectives for future FER studies. The contributions of our review can be summarized as follows: 1) Bridging the gap between advanced FER techniques and their practical utility in FER research, particularly in real-world unconstrained environment. Cutting-edge FER techniques have been difficult to access or use for researchers who may lack specific programming knowledge. Therefore, we provide an overview of newly developed, user-friendly FER toolboxes, thereby critically evaluating their neural architectures, training data, and performance. 2) Proposing new directions for future studies to utilize MLLMs for advancing the field of affective science: We provide an in-depth discussion that delves into the novel applications of MLLMs in FER research, shedding light on areas that have not received sufficient empirical attention. As outlined in section 3 of the manuscript, the utilization of MLLMs present exciting possibilities for advancing the field, such as analyzing contextual information to make robust emotion inferences for naturalistic facial behaviors.
- If you are afraid of your experiments, bring them so that you can be sure of the validity of the judgments of your method
Again, please note that our paper is a review article and not an empirical study. While we have successfully implemented and tested most of the mentioned toolboxes (with the exception of Affdex 2.0 and GPT-4V) on a small and privacy dataset, we were unable to conduct systematic experiments due to time constraints, the lengthy legal process for ethical approval, and limited computational resources at the present. We have referred to numerous empirical studies to support our claims about the potential utility of these advanced FER toolboxes and MLLMs in affective research. However, some arguments remain hypothetical due to the lack of concrete evidence, which we have acknowledged in the revised version of this paper to call for more empirical attention (p.11, 458-466).
- How to make sure that the emotions you recognized are true? Bring the evidence.
This issue has been one of the most challenging topics in investigating naturalistic facial expressions, specifically the interpretation of naturalistic expressions (p.2, lines 57-64). We are unable to verify the emotional experiences behind facial expressions sourced from the Internet using conversational approaches like self-report or EEG measurements. However, the emotions conveyed by facial expressions can be externally validated by observers. This validation method demonstrates a moderate level of agreement with subjective ratings of emotional experiences, as reported in studies on spontaneous facial expression recognition. This process is often referred to as inverse inference and can be achieved by training a model using facial expressions annotated with emotion labels by human observers (as in Py-Feat, see p.4, lines 155-157) or by referencing past research that documents the relationships between specific facial configurations and emotional states (as in Affdex 2.0, see p.4, lines 182-187). Furthermore, the validity of external ratings can be further enhanced through a comprehensive analysis of the congruent contexts in which facial expressions occur. These contexts can include the emotional stimuli (e.g., watching horrific movies), the conversations accompanied with facial expressions, the facial responses of interaction partners, etc. This is supported by the improved inter-rater agreement in emotion annotation when faces are presented with contextual cues (p.6, lines 269-272). Additionally, we have included new empirical evidence that demonstrates how analyzing the context can improve accuracy in emotion recognition (p.6, lines 311-324).
More importantly, it should be emphasized our main concern is not whether naturalistic facial expressions sourced from the Internet can genuinely reflect emotional states, as some expressions may be posed or enacted. Our primary focus is to examine the relationship between contexts and facial expressions (p.6, lines 253-262). Even if we were able to collect subjective ratings to validate the emotional experience of an expression as truly representing "happiness," we would still be unable to understand the meaning of "happiness" without taking the context into account (e.g., happiness as reflecting the anticipation of a music festival).
Round 2
Reviewer 3 Report
Comments and Suggestions for Authors
My comments are taken into account.